# Inhibition of Bacterial Efflux Pumps by Crude Extracts and Essential Oil from *Myristica fragrans* Houtt. (Nutmeg) Seeds against Methicillin-Resistant *Staphylococcus aureus*

**DOI:** 10.3390/molecules26154662

**Published:** 2021-07-31

**Authors:** Thidar Oo, Bhanubong Saiboonjan, Sukanya Srijampa, Arpasiri Srisrattakarn, Khaetthareeya Sutthanut, Ratree Tavichakorntrakool, Aroonwadee Chanawong, Aroonlug Lulitanond, Patcharaporn Tippayawat

**Affiliations:** 1Faculty of Associated Medical Sciences, Khon Kaen University, Khon Kaen 40002, Thailand; mt.thidaoo@gmail.com; 2Faculty of Medicine, Research Institute for Human High Performance and Health Promotion, Khon Kaen University, Khon Kaen 40002, Thailand; 3Center for Innovation and Standard for Medical Technology and Physical Therapy, Faculty of Associated Medical Sciences, Khon Kaen University, Khon Kaen 40002, Thailand; bhanubong.s@gmail.com; 4Biosensor Research Group for Non-Communicable Disease and Infectious Disease, Faculty of Associated Medical Sciences, Khon Kaen University, Khon Kaen 40002, Thailand; sukanya_s@kkumail.com (S.S.); arpasr@kkumail.com (A.S.); 5Faculty of Pharmaceutical Sciences, Khon Kaen University, Khon Kaen 40002, Thailand; khaesu@kku.ac.th; 6Centre for Research and Development of Medical Diagnostic Laboratories, Faculty of Associated Medical Sciences, Khon Kaen University, Khon Kaen 40002, Thailand; ratree.t@kku.ac.th (R.T.); aroonwad@kku.ac.th (A.C.); arolul@kku.ac.th (A.L.); 7Department of Medical Technology, Faculty of Associated Medical Sciences, Khon Kaen University, Khon Kaen 40002, Thailand

**Keywords:** nutmeg, crude extract, essential oil, methicillin-resistant *Staphylococcus aureus*, efflux pump inhibitor

## Abstract

*Myristica**fragrans* Houtt. (Nutmeg) is a widely known folk medicine across several parts of Asia, particularly used in antimicrobial treatment. Bacterial resistance involves the expression of efflux pump systems (chromosomal *nor*A and *mep*A) in methicillin-resistant *Staphylococcus aureus* (MRSA). Crude extract (CE) and essential oil (EO) obtained from nutmeg were applied as efflux pump inhibitors (EPIs), thereby enhancing the antimicrobial activity of the drugs they were used in. The major substances in CE and EO, which function as EPIs, in a descending order of % peak area include elemicin, myristicin, methoxyeugenol, myristicin, and asarone. Here, we investigated whether the low amount of CE and EO used as EPIs was sufficient to sensitize MRSA killing using the antibiotic ciprofloxacin, which acts as an efflux system. Interestingly, synergy between ciprofloxacin and CE or EO revealed the most significant viability of MRSA, depending on *nor*A and *mep*A, the latter being responsible for EPI function of EO. Therefore, CE and EO obtained from nutmeg can act as EPIs in combination with substances that act as efflux systems, thereby ensuring that the MRSA strain is susceptible to antibiotic treatment.

## 1. Introduction

*Myristica fragrans* Houtt. (nutmeg) is a well-known traditional medicine that is used in several countries, especially those in Asia [1]. Several in vitro and in vivo studies have attributed a wide array of pharmacological actions to nutmeg seeds, including antimicrobial [2], antioxidant [2,3], anti-inflammatory [2,4], antiparasitic [2], aphrodisiac [5], and hepatoprotective activities [6]. The pharmacological importance of nutmeg seed extract (essential oil) as an antimicrobial agent against Gram positive and Gram negative bacteria, and fungi is widely reported and is of great interest [6,7], particularly those of the main constituents of nutmeg including sabinene, 4-terpineol, myristicin, elemicin, tetradecanoic acid, α-pinene, β-pinene, β-phellandrene, methoxyeugenol, and γ-terpine, β-myrcene and their function as potential antimicrobial agents [6,7,8,9]. However, most of them found in essential oil of nutmeg seeds than those of the crude extracts of nutmeg seeds were mainly found to contain only methoxyeugenol [6]. In particular, EOs have been widely reported as disinfectant, with its major and minor components playing a role, as the antimicrobial chemical class control the EO’s mechanism against pathogens. They affect single or multiple targets within the pathogens. Therefore, the identification of the mode of action depends on the composition of the EOs [10].

World Health Organization (WHO) has reported antimicrobial resistant (AMR) Gram negative and Gram positive bacteria and has classified these microorganisms under the global priority pathogen list [11,12]. *Staphylococcus aureus* is an opportunistic pathogen and is associated with hospital acquired infections, serious nosocomial infections, and growing antimicrobial resistance against beta-lactams [13]. In addition, *S. aureus* species is among the largest global challenges faced by the clinical field, especially the methicillin-resistant *Staphylococcus aureus* (MRSA) strain [13,14].

The MRSA strain is of particular concern because of its ability to spread extensively and rapidly. Further, the multidrug resistance (MDR) of the MRSA strain to β-lactam and aminoglycoside antibiotics has led to serious problems with the treatment of their related infections, and this problem is increasing worldwide [13,15]. MRSA strains are associated with increased mortality in several common bloodstream infections, and their burden on diseases is poorly understood in many countries [16]. In Thailand, the presence of MRSA was documented in more than 20% of *S. aureus* strains between 2005 and 2006 [16]. Until now, the prevalence of MRSA infection in Thailand has been estimated to be 26%, with higher estimates (65%) among intensive care unit (ICU) patients [17].

Therefore, a better understanding of the current trends in MRSA strains is essential for their monitoring and control [16]. Further, the emergence of resistance in MRSA strain is a major threat [18]. Resistance mechanisms can also emerge due to mutations that alter the drug binding sites on molecular targets and increase the expression of endogenous efflux pumps [19]. Moreover, the bacterial efflux system is the main factor that causes MDR. Therefore, identifying effective efflux pump inhibitors that prevent the bacteria from expelling drugs out of the cells is a valid method of targeting bacteria [12].

The efflux systems serve an important role in antimicrobial resistance [19], hence, there have been several studies detailing efflux pump inhibitors (EPIs) [20]. An efflux pump belonging to the major facilitator superfamily (MFS) contributes to fluoroquinolones’ resistance, especially, ciprofloxacin [21]. There are more than 30 known efflux pump genes. Among them, *nor*A gene is the most studied efflux pump in *S. aureus*. Moreover, there have been studies concerning co-administration of efflux pump inhibitors with antibacterial drugs to overcome efflux-mediated resistance to the drugs by efflux NorA and MepA proteins [22]. Currently, compounds that have been identified as EPIs [21] tend to have fewer side effects compared with those of antibacterial drugs [23].

Nutmeg seeds are a natural compound that have been reported to exhibit antimicrobial, antioxidant, anticancer, and antiosteoporosis activities [23,24]. In addition, the properties of nutmeg crude extracts are known to inhibit MDR Gram negative bacteria [25]. Therefore, in the present study, we decided to study crude extract and essential oil of nutmeg seeds, and these in combination with ciprofloxacin against Gram positive bacteria such as MRSA strains. The study aimed to investigate efflux pump inhibitors of nutmeg crude extract and essential oil against the MRSA strains via efflux systems in in vitro culture.

## 2. Results

### 2.1. Determination of Chemical Composition of Crude Extract and Essential Oil

The seeds of nutmeg were extracted by two extraction methods. The yield of crude extract from the seeds using acetone extraction was 5.15% which was in pale-yellow color and sticky. The powdered seeds produced 2.47% yield which was oily and colorless.

Total phenolic and flavonoid contents of nutmeg seeds (dry powder) were determined using the colorimetric technique. The results for the total phenolic content (TPC) and total flavonoid content (TFC) of nutmeg crude extract (CE) and essential oil (EO) were as indicated by the mean ± SD in Table 1. The TPC, gallic acid equivalents (GAE)/100 g of dry weight was measured at 620 nm and the TFC, quercetin equivalent (QE)/100 g of dry weight was measured at 450 nm. The TPC was found to be 397 µg GAE/100 g of dry weight, which was more than TFC that was 21 µg QE/100 g of dry weight in CE. However, the TPC revealed 964 GAE/100 g of dry weight more than TFC which was 180 µg QE/100 g of dry weight in EO.

The identification of the chemical components in CE and EO was performed using gas chromatography mass spectrometry (GC/MS) analysis. In this study, GC/MS result of CE showed 45 chemical components and that with EO showed 44 chemical components (Appendix A). In the present study, we focused on the top 10 components found in CE and EO, as indicated by a high peak (% peak area) (Figure 1). In addition, top 10 representative natural compounds with the most of antimicrobial activity including elemicin, tetradecanoic acid, myristicin, 4-terpineol, sabinene, methoxyeugenol, cis-14-thujanol, decanoic acid, 2-oxo-, methyl ester, safrene, and n-hexadecanoic acid in CE; and sabinene, 4-terpineol, α-pinene, β-phellandrene, β-pinene, γ-terpinene, β-myrcene, myristicin, cis-asarone, and 1,3-benzodioxole, 5-(2-propenyl)- in EO. Sabine, 4-terpineol and myristicin were common components found in both CE and EO. Unique components in CE included cis-14-thujanol, safrene, elemicin, methoxyeugenol, tetradecanoic acid, n-hexadecanoic acid, decanoic acid, and 2-oxo-, methyl ester; and that in EO included α-pinene, β-pinene, β-myrcene, β-phellandrene, γ-terpinene, 1,3-benzodioxole, 5-(2-propenyl)-, and cis-asarone. The percentage of peak area correlation with antimicrobial activity alongside the name, molecular formula, and molecular weight of the compound are listed in Table 2. The components related to antimicrobial activity were investigated for antimicrobial activity against bacterial drug resistance, especially for methicillin-resistant *Staphylococcus aureus* (MRSA) via efflux pump mechanisms.

### 2.2. Determination of MRSA

This study explores the distribution pattern of antimicrobial activity of MRSA strains. The clinical strains included MRSA351, 352, 353, 354, and 355, and *S**. aureus* ATCC 25923, control strains were included and used for phenotypic antimicrobial susceptibility tests (AST). The five MRSA strains demonstrated resistance to cefoxitin and ciprofloxacin (CIP) (Table 3). However, *S**. aureus* ATCC 25923 strain was found to be susceptible to cefoxitin and ciprofloxacin (data not shown). Moreover, the genotypic test must generate to confirm resistant genes and efflux genes of MRSA strains by conventional PCR.

### 2.3. Genotypic Determination of Resistant Genes and Efflux Genes in MRSA Strains

Screening of resistant gene and efflux pump (EP) genes was conducted using the conventional PCR method. The results indicate the presence of the *mec*A (286 bp) gene in five MRSA strains. In addition, EP genes including *nor*A (436 bp) and *mep*A (91 bp) were detected in the five MRSA strains, which were represented using agarose gel electrophoresis (Appendix A). Five MRSA strains were found, both resistant genes and efflux pump genes. Furthermore, the expression levels of all strains including *nor*A and *mep*A, and chromosomal EP genes were determined.

The real-time PCR method was used for the detection and quantification of EP genes in MRSA strains. The expression levels of *nor*A and *mep*A in MRSA were upregulated and compared with those of *S**. aureus* ATCC 29213 and housekeeping genes (16S rRNA). The relative expression ratio of *nor*A gene in MRSA352, 353, 355, 351, and 354, indicated up-regulation (Figure 2A). Moreover, *mep*A genes showed a higher relative expression ratio of MRSA353 compared with that of MRSA351, 352, 354, and 355 (Figure 2B). The experiments were performed in triplicate and data were presented as mean ± SD. Therefore, the expression levels of both *nor*A and *mep*A in MRSA353 were found to be high, but the expression levels of both *nor*A and *mep*A in MRSA351 and 354 were found to be low.

### 2.4. Antimicrobial Susceptibility Testing

Antimicrobial susceptibility testing (AST) was conducted to know more about the details of chemical components involved in antimicrobial activity against five MRSA strains. Therefore, CE and EO were used for experimental testing in *S**. aureus* ATCC 29213 (MSSA) and five MRSA strains. CE against MRSA354 had minimal inhibitory concentration (MIC) and minimal bactericidal concentration (MBC) of 1250 µg/mL and 2500 µg/mL, respectively. CE against MRSA352 had a MIC and MBC of 78 µg/mL and 156 µg/mL, respectively. Moreover, the EO results showed that MIC and MBC values of MRSA355 were 0.195 and 0.391 of a percentage unit, respectively. However, MRSA351–354 exhibited similar MIC and MBC values of 0.098 and 0.195, respectively (Table 4). The MIC and MBC values of CE (µg/mL) in five MRSA strains showed significant differences of each strain while EO (% *v**/v*) presented the same value. In summary, CE and EO exhibited antimicrobial activity against five MRSA strains; however, MBC concentrations of both CE and EO were required up to 2-fold of that of MIC. Next we design-modified these data to evaluate synergistic effects of CE and EO with antibiotic (CE with CIP and EO with CIP).

### 2.5. Combination Activity of CE or EO with Ciprofloxacin

Synergistic combinations of CE or EO with CIP would be favorable to prevent drug resistance, especially against MRSA strain. The results are indicated in Table 5 which showed fractional inhibitory concentration (FIC) index range of >0.5 to 4 indicating additive/indifferent (A/I) effect. However, combined A/I effect of the extract and CIP could be indicated as synergistic activity following the interaction of natural compounds combined with antibiotic [26]. Thus, the CIP combined with CE or EO might provide a possibility of MRSA inhibition via efflux pump system, and the potential of efflux pump inhibitor (EPI) of the extracts would be investigated further.

### 2.6. Determination of Efflux Pump Inhibitors (EPIs)

Efflux pump inhibitors activity of CE and EO against the five MRSA strains was analyzed. We investigated this mechanism by phenotypic examination using agar disc diffusion, wherein the inhibition zone (mm) was observed in Figure 3A,B. The results showed that the antibacterial activity of combination of CIP and CE (CIP/CE) against MRSA351, 352, 353, and 354 with inhibition zones of 13.5 ± 0.4 mm, 11.8 ± 0.5 mm, 8.8 ± 1.6 mm, and 13.9 ± 0.1 mm, respectively, were significantly higher than CIP alone. The inhibition zones were significantly increased by CIP/EO combination against MRSA352 (10.8 ± 1.0 mm) and MRSA354 (10.2 ± 1.0 mm). Interestingly, the increasing inhibition zones of CIP/CE or CIP/EO combinations against various strains of MRSA indicated the potential of the extracts’ ability to control the pathogens with EPIs effect.

Intracellular accumulation of ethidium bromide (EtBr) inside the bacterial cells was shown in Figure 3C, indicated by the rate of efflux in different strains, which could be compared as follows: bacterial control or negative control (no treatment), the percentage of EtBr accumulation in the cells <3% of each MRSA control strains; and carbonyl cyanide m-chlorophenylhydrazone (CCCP) treatment or positive control, the percentage of EtBr accumulation in the cells >95% of each MRSA strains. The accumulation of EtBr inside the bacterial cells was significantly increased in the presence of CE and EO compared with the untreated. The CE treated-MRSA strains showed the EtBr accumulation from 34.6% to 79.7% and EO treatment presented the dye accumulation from 12.3% to 98.2%. Therefore, it could be indicated that CE and EO as EPIs were commonly referred to as multidrug or efflux transporters based on each MRSA strains via different efflux pump mechanisms used.

The CE and EO were used as EPIs in this study for five MRSA strains, and it was compared with CCCP, positive EPI control. The results found that the CIP/CE combination against MRSA351, 352, and 353 showed a significantly increased bacterial growth inhibition by 81.6% ± 9.4, 69.5% ± 6.8, and 67.5% ± 2.8 of inhibition, respectively, (Figure 4A) compared with CE treatment group, while there were no difference in MRSA354 and 355. However, this result would be indicated that the CE alone provided a high bactericidal ability without CIP combination. For EO study, the bacterial inhibition was significantly enhanced by CIP/EO combination against MRSA351 and 354 by 63.0% ± 3.4 and 97.5% ± 0.8 of inhibition, respectively, (Figure 4B) compared with EO alone. On the other hand, the EO alone presented high percentage of bacterial growth inhibition in MRSA352 and 355. According to these results, it would relate to the resistance genes expression and characteristics of the bacteria leading to the diversity response to the nutmeg extracts.

## 3. Discussions

Nutmeg or *Myristica fragrans* Houtt. is a well-known medicinal plant of interest for new drug development in tropical countries. A correlation has been indicated between the chemical compounds in nutmeg and antimicrobial activity against pathogenic infections [27]. Antimicrobial activity of nutmeg against Gram positive and Gram negative bacteria, and fungi has been reported. Nutmeg powder possess several active phytochemicals, which include phenols and flavonoids [6,24]. The content of phenols (3.97 µg GAE/g) was higher than that of flavonoids (0.21 µg QE/g) in crude extracts (CE) and the content of phenols (9.64 µg GAE/g) was higher than that of flavonoids (1.80 µg QE/g) in essential oils (EO). The content of phenols and flavonoids was very low in nutmeg (seed powder); however, EO has phenols and flavonoids contents much higher than the CE. The total content of phenols and flavonoids were found to be affected by solvent or method used for extraction [24]. EO was found to be notable for extraction of phenolic and flavonoid compounds than CE used in this study in accordance with a previous study done in 2013 [28].

Recently, several studies have indicated the presence of antimicrobial properties in EO of nutmeg seeds [9,29,30]. However, there are fewer studies detailing CE compared with those using EO [29]. Therefore, in the present study, we focused on CE and EO in nutmeg seeds using GC/MS analysis. The results found that nutmeg seeds revealed common or unique elements in CE and EO. We focused on three main compounds found in CE and EO, namely sabinene, 4-terpineol, and myristicin. It has been reported that these compounds exhibit antimicrobial activity against pathogenic bacteria (Gram positive and negative) and fungi [6,9,24,29,30,31,32]. Moreover, unique compounds including cis-14-thujanol, safrene, elemicin, methoxyeugenol, tetradecanoic acid, n-hexadecanoic acid, decanoic acid, and 2-oxo-, methyl ester in CE; and α-pinene, β-pinene, β-myrcene, β-phellandrene, γ-terpinene, 1,3-benzodioxole, 5-(2-propenyl)-, and cis-sarone in EO have been reported to exhibit antimicrobial activity in nutmeg and other plants [33,34]. Although it is important to analyze all the compounds in CE (total of 45) and EO (total of 44), our study considered only the top 10 compounds, which showed a high percentage of peak area and exhibited antimicrobial activity. The importance of methoxy group has been reported as efflux pump inhibitors (EPI) via efflux transporter NorA protein in *S. aureus* [35]. The synthesis of compounds carrying methoxy groups seems to be of great interest; therefore, a number of known compounds have been reported from this study including myristicin, elemicin, and methoxyeugeno in CE, and myristicin and cis-asarone in EO. These compounds were EPIs in MRSA via *nor*A and *mep*A. The molecular structure of four compounds such as elemicin, myristicin, methoxyeugenol, and cis (β)-asarone are shown in Figure 5.

The MRSA is a major health concern that causes a wide range of infections globally [36]. The success of MRSA infections is due to an array of virulence factors, genetic flexibility, and multidrug efflux transport proteins, and these have allowed MRSA to develop resistance to a wide range of antibiotics [37]. A key role to fluoroquinolone (FQ) resistance in MRSA is the efflux pump (EP) mechanism; such combination may have a great impact on resistance to ciprofloxacin (CIP) [38]. Previous studies have reported that EP plays an important role in virulence, and a lack of EP or defects in EP were closely associated with attenuated virulence [37]. The *norA* and mepA genes, encoding the transporter proteins, are located on the chromosome [39]. The major facilitator superfamily (MFS), *nor*A, is a key modulator of antimicrobial resistance in S. aureus, and it has been established using whole genome sequences and the function of EP has been characterized. However, multiple antimicrobial extrusion protein family (MATE), *mep*A, confers low-level resistance to FQ, such as CIP [39]. FQ resistance in S. aureus may be mediated by the *nor*A- and *mep*A-encoded fluoroquinolones efflux pump systems [40]. In the present study, we aimed to evaluate efflux pump genes *nor*A and *mep*A, which are correlated to *mec*A, the MRSA detectable gene [41]. The results indicated that EPs were key modulators of antimicrobial resistance and had a critical role to play against *S. aureus*.

Several families of proteins capable of multiple drug extrusion have been described. Primary transporters require ATP hydrolysis for drug transport (ATP binding cassette, or ABC pumps), and secondary transporters require a proton or sodium gradient for drug efflux (MFS- and MATE-type pumps) [22]. These pumps can extrude multiple structurally unrelated compounds that lead to MDR phenotype [22]. FQ resistance in S. aureus was mediated by the *nor*A- and *mep*A-encoded FQ EP systems [22]. These pump systems recognize FQs and provide, depending on the expression levels, efflux-mediated resistance to many FQs [22]. Therefore, inhibition of EPs has also been shown to suppress the emergence of higher-level resistance mechanisms [22]. Methoxyl groups have been reported to be EPIs, the potentiality of compounds *3b* and *7d*, two novel methoxy-2-phenylquinoline derivatives to inhibit NorA efflux has been proved by Felicetti et al. in 2018. Both compounds exhibited depolarizing effect at their minimal potentiating concentration 8-fold CIP MIC values on the *S. aureus* membrane, thereby excluding a NorA efflux inhibition due to the proton motive force disruption. These data suggest that compounds *3b* and *7d* may inhibit NorA efflux mechanism in a specific manner such as proton motive force disruption, competitive extrusion with efflux drugs, or increase in membrane permeability [35,42]. Moreover, both compounds can inhibit the MepA efflux pump. These data revealed methoxy-2-phenylquinoline scaffold as suitable to obtain broad spectrum EPIs that are strongly needed to fully overcome a pump-mediated resistance [35]. Structure modifications of 2-Phenylquinolone were necessary to obtain two series *3b* and *7d*, two novel methoxy-2-Phenylquinolone derivatives that were reported from previous study [22], and the picture was generated by using ChemDraw pro 8.0 (Figure 6).

The EPI activity may be related to the EP gene expression. The inhibition zone of CIP against MRSA351 and MRSA354 increased in MHA containing CE and EO, as it had lower levels of *nor*A and *mep*A than those did other strains. In addition, MRSA352 strain was increasingly inhibited around the drug disc on MHA with the extracts; however, MRSA352 demonstrated a high level of *nor*A expression. Based on these results, it can be assumed that CE and EO may inhibit NorA function by means of the major compounds, which have methoxyl group, such as elemicin, myristicin, and methoxyeugenol in CE, and myristicin and cis-asarone in EO. The methoxylated compounds have been reported as an important feature of potential EPIs [35]. Methoxyl groups increased the lipophilicity and modulatory effect on FQ resistance in *S. aureus* SA1199-B [43]. The membrane intercalation of lipophilic compounds could inhibit efflux systems, which used the proton motive force to drive drug transport, such as NorA, leading to a higher intracellular accumulation of antibiotics in the bacterial cell [44]. Furthermore, significant polarity (hydrophobicity) was necessary to guarantee affinity of a compound with EPs, including NorA [45]. Similarly, a previous study has reported that phytochemical in *Arrabidaea brachypoda*, as Brachydin B, was able to potentiate the activity of norfloxacin against *S. aureus* overexpressing *nor*A [46]. However, MRSA353 *mep*A overexpression was not inhibited by EO, possibly, as the EO compounds were not specific to MepA, which led to the drug being pumped out of the bacterial cell.

Concerning the extracts presenting with EPIs, the authors determined that EPI ability enabled CIP to have the maximum efficacy. Thus, the concentration of the extracts was reduced to decrease the effects of the extracts on the inhibition of bacteria. The CE possessed EPI ability and enhanced efficacy ciprofloxacin to inhibit bacterial growth of MRSA351, 352, and 353 by 80%. EO enhanced the efficacy of ciprofloxacin to inhibit growth of MRSA351 and MRSA354 by 60% and 98%, respectively. Moreover, CE and EO could function directly to inhibit MRSA354, MRSA352, and MRSA355 without combination effect. This was supposed to contribute from their major compounds in the extracts, which showed antimicrobial activity, including sabinene, 4-terpineol, myristicin, elemicin, tetradecanoic acid, α-pinene, β-pinene, and β-phellandrene [6,7,8,9]. Thileepan et al. reported that the extract of M. fragrans seeds showed antimicrobial activity against S. aureus and MRSA strains [47]. Interestingly, CE and EO did not enhance ciprofloxacin activity against MRSA355; however, the extracts could possibly possess antimicrobial ability. Antibiotic resistance could be attributed to different mechanisms, including lower antibiotic influx, enzymatic inactivation of the antibiotic, and mutations that reduce the affinity of the antibiotic on its target [48].

## 4. Materials and Methods

### 4.1. Preparation of Crude Extract and Essential Oil

Nutmeg seeds were collected from a local traditional market of Mandalay in Myanmar. To prepare nutmeg crude plant extracts, the seeds (dry powder) were extracted using a maceration technique previously described by Gupta et al. [24]. Essential oil was extracted using the hydro-steam distillation method as previously described by Al-Mariri and Safi [49]. Post extraction of essential oil, we proceeded with the isolation and purification of bioactive compounds from the nutmeg extracted using the Folin–Ciocalteu method, as previously described by Lahmar et al. [50]. Additional details concerning the total phenolic content and total flavonoid content [51] are given in the Appendix A and methods. Gas chromatography mass spectrometry (GC/MS) was performed as described by Matulyte et al. [52]. The bacteria stocks (five MRSA strains and S. aureus ATCC 25923) were cultured on MHA plate and incubated at 37 °C for 24 h. A single colony of bacteria was picked and inoculated in nutrient broth (Himedia, Mumbai, India) under aerobic conditions at 37 °C for 18 h (midlog-phase of bacterial growth). Then the bacteria were washed and adjusted to 0.5 McFarland (10^8^ CFU/mL) by 0.85% normal saline.

### 4.2. Bacterial Strains

Bacterial strains used in this study included *Staphylococcus*
*aureus* (*S**. aureus* ATCC 29213 and 25923), a reference strains, and five methicillin-resistant *S**. aureus* (MRSA), clinical strains that were obtained from the Department of Clinical Microbiology, Faculty of Associated Medical Sciences, Khon Kaen University, Thailand. Five strains of MRSA have been published previously by Sutthamee et al. in 2019 [53].

### 4.3. Antimicrobial Susceptibility Testing

The standardized disc diffusion method can be used commonly for susceptibility testing. The disc diffusion test was performed using Mueller Hinton agar (Himedia, Mumbai, India) with cefoxitin (30 µg) and ciprofloxacin (5 µg) discs (Oxoid Ltd.; Basingstoke, Hampshire, UK). The bacterial suspension of 1 × 10^8^ CFU/mL of *S. aureus* ATCC 25923 and five MRSA strains were spread on MHA plates. The drug discs were placed over the plates, followed by incubation at 37 °C for 24 h.

Minimal inhibitory concentration (MIC) and minimal bactericidal concentration (MBC) testing was conducted. Susceptibility to nutmeg crude and essential oil extractions were determined by broth microdilution methods performed using Mueller Hinton broth (Himedia, Mumbai, India). Briefly, the extractions were diluted using twofold serial dilutions from 5000 to 0.5 µg/mL of CE and 50 to 0.0005% of EO. Then, bacterial suspension of 1 × 10^8^ CFU/mL of *S. aureus* ATCC 29213 and five MRSA strains were added in the wells, followed by incubation at 37 °C for 24 h. The experiments were performed according to the Clinical & Laboratory Standards Institute (CLSI) recommendations.

### 4.4. Polymerase Chain Reaction (PCR) Method

The microorganism strains were cultivated on Muller Hinton agar (Himedia, Mumbai, India), incubated at 37 °C for 24 h. Further, the boiling method was used for DNA extraction in this study [54]. Primers used in this study are follows: *nor*A sequence (5′-->3′) 436 bp, forward: GTTACTTGTTGCTGCTTTTG and reverse: GCTTGTCGTAGACTTTTTCG [55] and *mec*A sequence (5′-->3′) 286 bp, forward: TGCTATCCACCCTCAAACAGG and reverse: AACGTTGTAACCACCCCAAGA [54]. PCR amplification was performed using FlexCycler2 PCR thermo cycles machine (Analytik Jena AG, Thuringia, Germany). Amplification was achieved by a series of steps: *nor*A and *mep*A genes were initially denatured at 94 °C for 5 min; 35 cycles of 94 °C for 30 s, 58 °C for 30 s, 72 °C for 30 s, followed by final elongation at 72 °C for 7 min. For the *mec*A gene, the following temperature cycling was used: initial denaturing at 95 °C for 5 min, 30 cycles of 95 °C for 1 min, 55 °C for 1 min, 72 °C for 1 min, and a final extension at 72 °C for 10 min. Analysis of PCR products was conducted using 2% agarose gel electrophoresis (Cleaver Scientific, Warwickshire, UK). The gels were then exposed to UV light and the images were generated using the GeneFlash gel documentation system (Syngene, Frederick, MD, USA).

### 4.5. RNA Extraction and cDNA Synthesis

Bacterial cells were harvested at 24 h of incubation by centrifugation at 4 °C at 4500× *g* for 30 min. The supernatant was then discarded. RNA was strained using 1 mL of Trizol reagent (Invitrogen, Waltham, MA, USA) per reaction. The cell lysate was mixed with 200 µL of chloroform, shaken, and kept at 25 °C for 2 min. Centrifugation was then conducted at 12,000× *g* for 30 min at 4 °C. Further, the upper aqueous phase was carefully transferred into a new microcentrifuge tube. Further, 500 µL of isopropanol was added, shaken, and kept at room temperature for 10 min. This was followed by centrifugation at 12,000× *g* for 30 min at 4 °C. The supernatant was then discarded. Further, washing was conducted using 75% *v/v* ethanol at 1 mL per reaction. The ethanol was then removed, followed by air-drying (20 min). The solution was then redissolved using 50 µL of 0.25% DEPC. RNA concentration was determined using spectrophotometry at 260 nm.

cDNA was synthesized using MaximeTM RT PreMix (iNtRON Biotechnology, Gyeonggi-do, Korea). The total reaction volume was 20 µL per reaction. Briefly, 1 µg of RNA template and sterile distilled water were added into the Maxime RT PreMix tubes up to 20 µL. cDNA synthesis was then performed as follows: reaction steps of cDNA synthesis at 45 °C for 60 min, followed by RTase inactivation at 95 °C for 5 min. Then, 50 µL of sterile distilled water was added into the tubes. cDNA concentration was determined using spectrophotometry at 260 nm.

### 4.6. Quantitative Real-Time PCR (qPCR) Method

Primers were synthesized by Ward Medic, Thailand. Primers used in this study are follows: *nor*A sequence (5′—>3′) 436 bp, forward: GTTACTTGTTGCTGCTTTTG and reverse: GCTTGTCGTAGACTTTTTCG [55]; *mep*A sequence (5′—>3′) 91 bp, forward: TTATGGAAACTTCGCGATTGC and reverse: AACACCTTCACATAATCCCATGATAAT [56] and 16S rRNA sequence (5′—>3′) 1478 bp, forward: CCTGGCTCAGGATGAACG and reverse: AATCATTTGTCCCACCTTCG [57]. 16S rRNA was determined as the housekeeping gene, by referring to a previous study. qPCR was performed using Maxima SYBR Green/ROX qPCR Master Mix (2X) (Thermo Scientific™, Waltham, MA, USA). qPCR and analysis were performed using a real-time PCR instrument (Applied Biosystems QuantStudio 6 Flex Real-Time PCR System). Primer annealing temperatures was 58 °C for all primer pairs. The amplification products obtained were confirmed by used a specific melting curve analysis. The relative expression of genes was calculated using the 2^−^^∆∆Ct^ formula with 16S rRNA as the housekeeping gene, along with *S**. aureus* ATCC 29213 growth control.

### 4.7. Checkerboard Titration Assay

The combination of ciprofloxacin (CIP) and CE or EO was assessed using a checkerboard titration assay. The CIP was examined at seven concentrations (0.25 to 16 µg/mL), while CE and EO were examined at 7 concentrations (0.156 to 10 µg/mL) and (0.09 to 6.25% *v/v*), respectively. The determination of the association between antibiotics and natural compounds were performed using FIC index where an FIC ≤ 0.5 indicates synergistic, >0.5 to ≤1 indicates additive, 1 to 4 indicates indifferent, and >4 indicates antagonistic and is interpreted as A/I, Additive/Indifferent. The experiments were performed according to Gradelski et al. in 2001 [26].

### 4.8. Examination of Inhibition Zone, CE and EO with Ciprofloxacin Combinations by Agar Disc Diffusion

Agar disc diffusion was conducted with CE, EO, and 10 µg/mL of carbonyl cyanide m-chlorophenylhydrazone (CCCP) (Sigma-Aldrich, Saint-Quentin-Fallavier, France) on Mueller Hinton agar (MHA) plates. Final concentration of CE used was 310, 750, 620, 1250, and 310 µg/mL for MRSA 351, 352, 353, 354, and 355, respectively. Then, 0.098% EO was used for MRSA 351 to 354 strains and 0.195% EO was used for MRSA355. Then, 0.5 McFarland units (10^8^ CFU/mL) of five MRSA strains were used as culture condition on an agar plate with CE, EO, and CCCP, followed by incubation at 37 °C for 24 h. After that, the zone diameter was measured. The experiments were then performed in triplicate.

### 4.9. Evaluation of Ethidium Bromide Accumulation in the MRSA Strains by Flow Cytometry

Five MRSA strains were cultured using 10 mL of nutrient broth medium, followed by incubation at 37 °C for overnight. Bacterial cells were harvested by centrifugation at 10,000 rpm for 10 min, following which the supernatant was discarded. The cells were then washed twice with phosphate buffer saline (PBS). Bacterial O.D. was equal to 0.3 in PBS, and 3 µg/mL (final concentration) of Ethidium bromide (EtBr) were added to the test tube. The CCCP (20 μg/mL) was used to as positive control. Final concentration of CE used was 310, 750, 620, 1250 and 310 µg/mL for MRSA 351, 352, 353, 354, and 355, respectively. Final concentration of EO used was 0.09% for MRSA 351, 352, 353, and 354 strains and 0.19% for MRSA355. The total volume, 0.5 mL was incubated at 37 °C for 15 min. The EtBr accumulation in the cells was measured by the fluorescence detected through 585 nm filter (FL-2 channel) using the FACSCanto II flow cytometer (BD Biosciences, San Jose, CA, USA). Data were collected for at least 10,000 events per sample [58]. The experiments were performed in triplicate.

### 4.10. Inhibition of Bacterial Growth by CE and EO with Ciprofloxacin Combinations

Five MRSA strains were prepared and adjusted to 10^5^ CFU/mL. The concentration of CIP used was 4 µg/mL (final concentration), while the concentration of CCCP used was 20 µg/mL. Final concentration of CE used was 150, 75, 310, 310, and 150 µg/mL for MRSA351, 352, 353, 354 and 355, respectively. The final concentration of EO used was 0.024% for MRSA351, 352, 353, and 354 and 0.048% *v**/v* for MRSA355, which was added into the bacterial tube. This was followed by incubation at 37 °C. Bacterial growth was assessed using colony count technique. The experiments were performed in triplicate.

## 5. Conclusions

Nutmeg is used for new drug development, especially for antimicrobial treatment owing to its various pharmacological activities. The study aims to focus on the activity of CE and EO obtained from nutmeg against five strains of MRSA with CIP resistance via EP mechanisms. The components from CE and EO were analyzed, and we found that most of these were related to antimicrobial properties. Especially, the content of methoxyl groups were reported to possess EPI activity and the combination of natural compound with CIP showed synergistic effect functionally facilitated by CIP through EP mechanisms. Therefore, EPI has potential as a new therapeutic agent, especially since it uses natural compounds such a nutmeg (CE and EO) as an alternative treatment. There is a need for a combination trial focused on EPI in the future, to aid the development of new drugs that have beneficial effects. Interestingly, antimicrobial activity against MRSA infection via EP mechanisms and other microbial infections is recommended to be the area of focus for new drugs development.

## Figures and Tables

**Figure 1 molecules-26-04662-f001:**
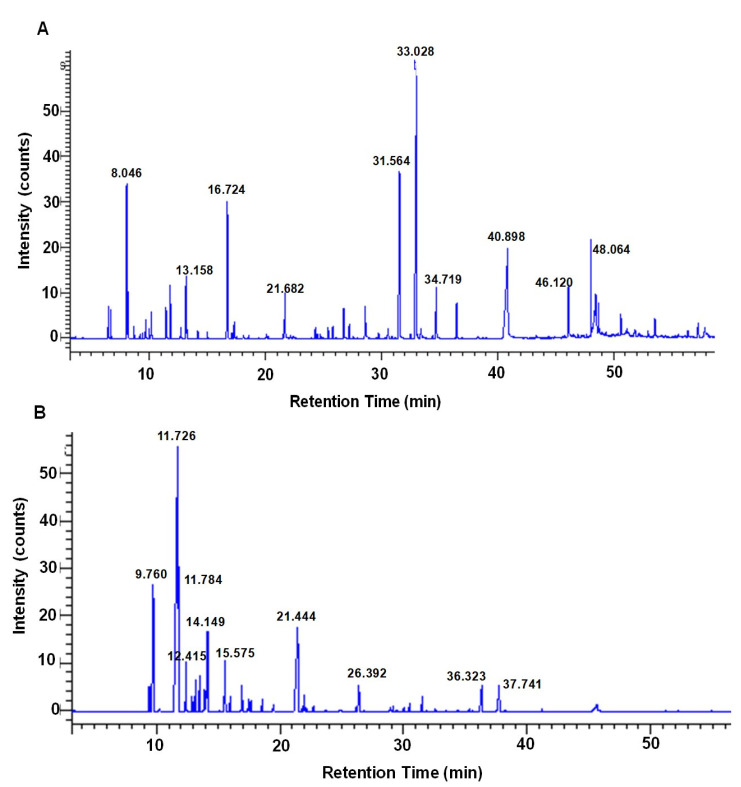
GC/MS analysis chromatogram of nutmeg. The major compounds are indicated in the high peak area (%) representative chromatogram, with 0 to 60 min retention time-8.046 (sabinene), 13.158 (cis-14-thujanol), 16.724 (4-terpineol), 21.682 (safrene), 31.564 (myristicin), 33.028 (elemicin), 34.719 (methoxyeugenol), 40.898 (tetradecanoic acid), 46.120 (n-hexadecanoic acid) and 48.064 (decanoic acid, 2-oxo-, methyl ester)-obtained from crude extract (**A**), and 9.760 (α-pinene), 11.726 (sabinene), 11.784 (β-pinene), 12.415 (β-myrcene), 14.149 (β-phellandrene), 15.575 (γ-terpinene), 21.444 (4-terpineol), 26.392 (1,3-benzodioxole, 5-(2-propenyl)-, 36.323 (myristicin) and 37.741 (cis-asarone)-obtained from essential oil (**B**).

**Figure 2 molecules-26-04662-f002:**
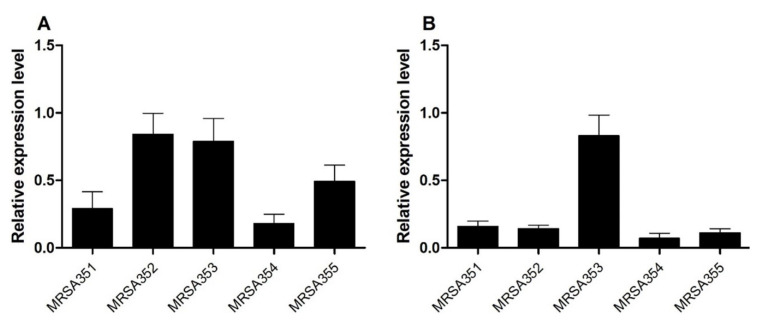
The relative gene expression of the EP genes. The levels of *nor*A (**A**) and *mep*A (**B**) in the five MRSA strains were performed by qPCR examining differential modulation of mRNA. Relative expression ratio (fold-change) represents the MRSA associated with EP genes at 24 h, which was calculated using the 2^−ΔΔCT^ method. All experiments were performed in triplicate. The data are presented as the mean of expression fold-change normalized against the expression level of *S. aureus* ATCC 29213 and 16S rRNA.

**Figure 3 molecules-26-04662-f003:**
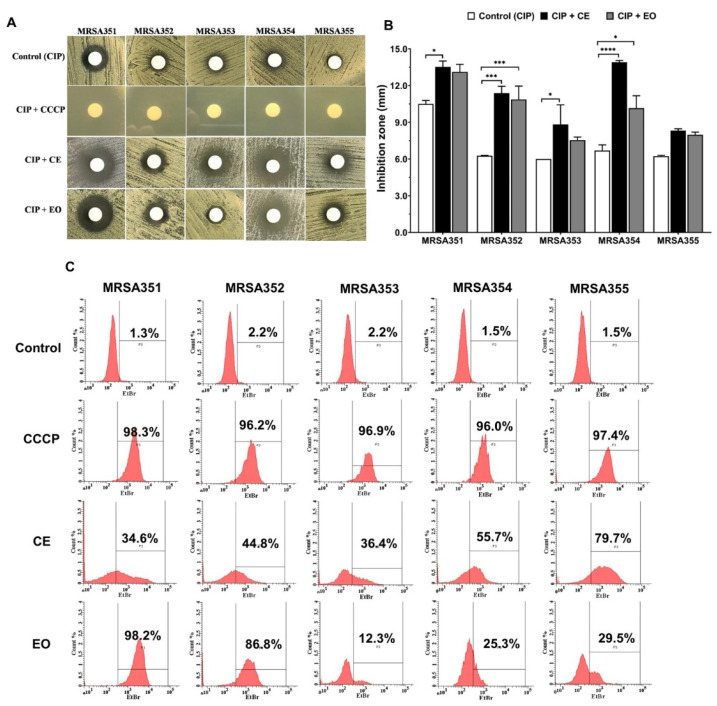
Potential effects of nutmeg crude extract (CE) and essential oil (EO) as efflux pump inhibitors in the five MRSA strains. Agar disc diffusion of the MRSA, combination of ciprofloxacin (CIP) with CE or EO, and its comparison with CIP and CIP with carbonyl cyanide m-chlorophenylhydrazone (CCCP) (**A**); inhibition zone (mm) of the MRSA, CIP with EC or EO, compared with CIP alone (**B**); flow cytometer analysis of intracellular accumulation inside the bacterial cells as indicated by the percentage of accumulation, including the control (EtBr), CCCP with EtBr, CE with EtBr, and EO with EtBr (**C**). *: *p* < 0.05; ***: *p* < 0.001; ****: *p* < 0.0001.

**Figure 4 molecules-26-04662-f004:**
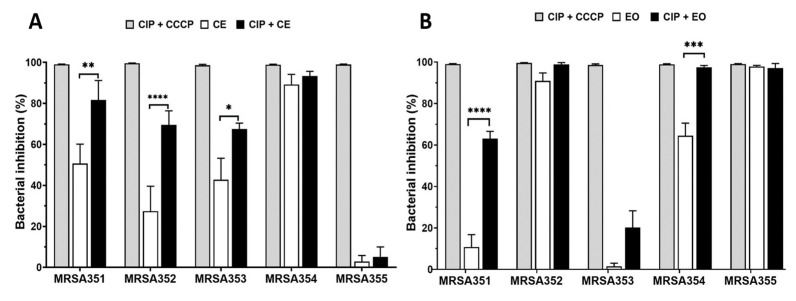
The potential effects of nutmeg crude extract (CE) and essential oil (EO) with CIP against five MRSA strains. The effects of CCCP, CE, and CE with CIP combination (**A**), and CCCP, EO, and EO with CIP combination (**B**) were determined by analyzing the percent inhibition of bacterial growth (colony counting method). *: *p* < 0.05; **: *p* < 0.01; ***: *p* < 0.001; **** *p* < 0.0001.

**Figure 5 molecules-26-04662-f005:**
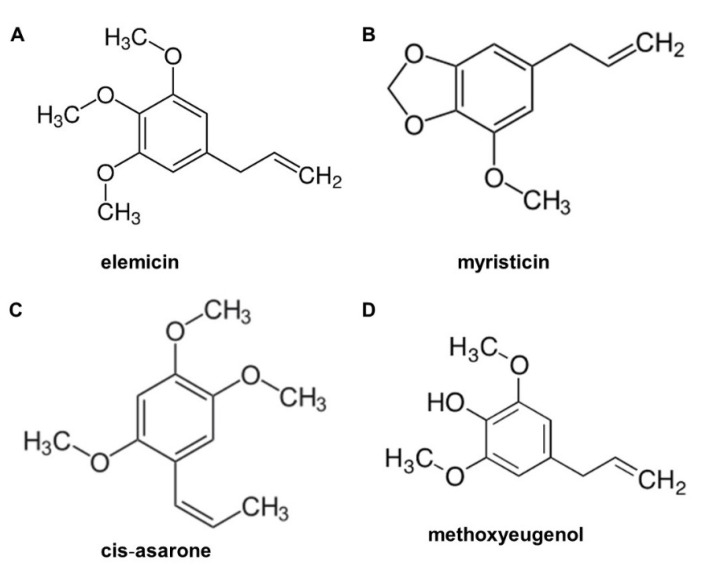
Molecular structure of active compounds found in nutmeg extractions. (**A**) elemicin, (**B**) myristicin, (**C**) cis-asarone, and (**D**) methoxyeugenol.

**Figure 6 molecules-26-04662-f006:**
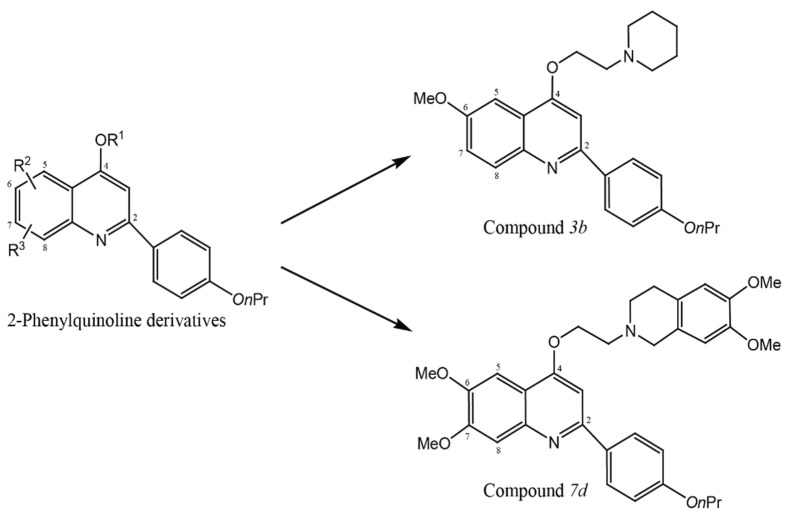
Structure modifications of 2-Phenylquinolone derivatives were required to obtain two series, two novel methoxy-2-Phenylquinolone derivatives (*3b* and *7d*). The picture was generated by used ChemDraw pro 8.0.

**Table 1 molecules-26-04662-t001:** The total phenolic and flavonoid contents of crude extract and essential oil.

Nutmeg (Seed Powder)	TPC ^a^ (µg)	TFC ^b^ (µg)
Crude extract	397 ± 0.70	21 ± 0.08
Essential oil	964 ± 0.98	180 ± 0.26

TPC: total phenolic content, TFC: total flavonoid content; ^a^ Gallic acid equivalents (GAE)/100 g of dry weight was measured at 620 nm; ^b^ Quercetin equivalent (QE)/100 g of dry weight was measured at 450 nm.

**Table 2 molecules-26-04662-t002:** Chemical composition in the crude extract and essential oil identified by GC/MS.

Peak Area (%)	Molecular Formula	Peak Name	RT ^a^ (min)	MW ^b^ (g/mol)
**Crude extract**				
22.663	C_12_H_16_O_3_	Elemicin	33.028	22.66
15.234	C_14_H_28_O_2_	Tetradecanoic acid	40.898	228.37
11.183	C_11_H_12_O_3_	Myristicin	31.564	192.21
6.880	C_10_H_18_O	4-terpineol	16.724	154.25
5.771	C_10_H_16_	Sabinene	8.046	136.23
2.774	C_11_H_14_O_3_	Methoxyeugenol	34.719	194.22
2.740	C_10_H_18_O	cis-4-Thujanol	13.158	154.25
2.603	C_4_H_6_O	Decanoic acid, 2-oxo-, methyl ester	48.064	200.27
2.283	C_10_H_10_O_2_	Safrene	21.682	162.18
2.283	C_16_H_32_O	n-Hexadecanoic acid	46.120	256.42
**Essential oil**				
36.907	C_10_H_16_	Sabinene	11.726	136.23
11.544	C_10_H_18_O	4-terpineol	21.444	154.25
9.414	C_10_H_16_	α-pinene	9.760	136.23
6.135	C_10_H_16_	β-phellandrene	14.149	136.23
4.445	C_10_H_16_	β-pinene	11.784	136.2
3.312	C_10_H_16_	γ-terpinene	15.575	136.23
2.624	C_10_H_16_	β-myrcene	12.415	136.23
2.551	C_11_H_12_O_3_	Myristicin	36.323	192.21
2.549	C_12_H_16_O_3_	Cis-asarone	37.741	208.25
1.884	C_10_H_10_O_2_	1,3-Benzodioxole, 5-(2-propenyl)-	26.392	162.18

^a^ RT, retention time; ^b^ MW, molecular weight.

**Table 3 molecules-26-04662-t003:** Detection of MRSA strains with cefoxitin and ciprofloxacin.

Microorganism	Cefoxitin		Ciprofloxacin	
	Inhibition Zone (mm)	Interpretation *	Inhibition Zone (mm)	Interpretation *
MRSA351	6	^a^R	8	R
MRSA352	13	R	6	R
MRSA353	6	R	6	R
MRSA354	6	R	6	R
MRSA355	6	R	6	R

* Clinical & Laboratory Standards Institute (CLSI) used as the standard and for the guidelines interpretation of antimicrobial susceptibility testing (AST) in the experiments; ^a^ R, resistant.

**Table 4 molecules-26-04662-t004:** Determination of MIC and MBC values using crude extract (CE) and essential oil (EO) against the MRSA.

Microorganism	CE (µg/mL)		EO (% *v**/v*)	
	MIC ^a^	MBC ^b^	MIC	MBC
MRSA351	312	625	0.098	0.195
MRSA352	78	156	0.098	0.195
MRSA353	625	1250	0.098	0.195
MRSA354	1250	2500	0.098	0.195
MRSA355	312	625	0.195	0.391

^a^ MIC, minimal inhibitory concentration; ^b^ MBC, minimal bactericidal concentration.

**Table 5 molecules-26-04662-t005:** The combination effects of ciprofloxacin with CE or EO against the MRSA.

Microorganism	∑FIC of CE	Interpretation *	∑FIC of EO	Interpretation
MRSA351	3.40	A/I ^a^	0.98	A/I
MRSA352	1.41	A/I	2.62	A/I
MRSA353	1.37	A/I	1.05	A/I
MRSA354	0.88	A/I	1.23	A/I
MRSA355	1.34	A/I	1.90	A/I

***** Fractional inhibitory concentration (FIC) index of ≤0.5 indicates synergistic, >0.5 to ≤1 indicates additive, 1 to 4 indicates indifferent, and >4 indicates antagonistic [26]; ^a^ A/I, Additive/Indifferent.

## Data Availability

Data of the compounds are available from the authors upon reasonable request.

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
