# Peer review of "Inhibition of Bacterial Efflux Pumps by Crude Extracts and Essential Oil from Myristica fragrans Houtt. (Nutmeg) Seeds against Methicillin-Resistant Staphylococcus aureus"

_molecules, 2021, doi:10.3390/molecules26154662_

Round 1

Reviewer 1 Report

Comments for Authors:

The manuscript is a well-conducted study on an important topic combatting antibiotic resistant pathogenic bacteria. As bacterial resistance involves the expression of efflux pump systems (chromosomal norA and mepA) in methicillin-resistant Staphylococcus aureus (MRSA), crude extract (CE) and essential oil (EO) from nutmeg were applied as efflux pump inhibitors (EPIs), to enhance the antimicrobial activity of the drugs they are to be used with.

Detailed comments:

  1. Page 13, Line 403: Under 4. Materials and Methods, 4.2. Bacterial strains:

Please describe how the bacteria were grown, at which growth phase they were harvested (if used at 108 CFU/mL concentration), were they fresh midlog-phase cells or diluted stationary cultures?

As later under 4.3. Antimicrobial susceptibility testing it is mentioned that “A total of 0.5 McFarland units (108 CFU/mL) of S. aureus 403 ATCC 25923 and five MRSA strains were used as culture condition, followed by incubation at 37 °C for 24 h”, does it mean that an overnight liquid culture was used for the plating, or an aliquot of a mid-log phase fresh culture at 108 CFU/mL was spread on Mueller-Hinton (M-H) agar plates with discs and incubated overnight?

Overnight cultures with stationary phase bacterial cells consist of cells of different ages, with vulnerable young and aged cells, that will exhibit different responses to the antimicrobials.

  1. Page 8, Line 200: Although the antimicrobial effect of the nutmeg components is confirmed, however, could the Authors add some comments regarding the practical application of these results? Based on the data in Table 4, of the crude extract 78 – 1250 µg/mL concentrations are required for minimal inhibitory concentration (MIC), and even more, up to 2500 µg/mL for minimal bactericidal concentration (MBC). Even if microliter volumes were added to the discs on the M-H agar, these are significant amounts of a natural compound with unknown immune response to be used in medical treatments.

Would these nutmeg compounds be applied to treat skin infections? If CE or EO are mixed with ciprofloxacin (CIP), as suggested for synergistic effect on Page 8, would they be administered as an injection or orally? If so, could any adverse reaction be expected from the patient’s immune system?

Author Response

Thank you for your suggestions. The authors have added and revised the information as your comments.
Please see the attachment.

Reviewer 2 Report

Nutmeg is the kernel of the seed within the fruit of Myristica fragrans and has several pharmacological activities, including antibacterial, in the present study the crude extract and essential oil has inhibitory activity of bacterial efflux pumps again MRSA, this is a well developed study with novel results. I have a few comments, as follows:

1) Expand information on the antimicrobial activities of essential oils: In the present study, crude extract and essential oils obtained from nutmeg are applied as efflux pump inhibitors but in the introduction there is no information about essential oils that are found naturally in plants and have long been used as antimicrobials, but most have not been investigated. This study explores essential oils as alternative antimicrobials to evaluate their ability to contribute to the control of antimicrobial resistant, therefore, it is important to investigate the potential of these essential oils, in particular for the disinfection in clinical and non- clinical environments.

Therefore, it is necessary for the authors to expand the information on the antimicrobial activities of essential oils, see below:

Khaled A. El-Tarabily, Mohamed T. El-Saadony, Mahmoud Alagawany, Muhammad Arif, Gaber E. Batiha, Asmaa F. Khafaga, Hamada A.M. Elwan, Shaaban S. Elnesr, Mohamed E. Abd El-Hack (2021). Using essential oils to overcome bacterial biofilm formation and their antimicrobial resistance. Saudi Journal of Biological Sciences, In press https://doi.org/10.1016/j.sjbs.2021.05.033.

2) Update Introduction and References: There are new articles on the essential oil of nutmeg (Myristica fragrans) that report its antimicrobial, antiseptic, antiparasitic, anti-inflammatory and antioxidant properties and that are not included in the background, so it is necessary to update the introduction, see below:

Matulyte I, Jekabsone A, Jankauskaite L, Zavistanaviciute P, Sakiene V, Bartkiene E, Ruzauskas M, Kopustinskiene DM, Santini A, Bernatoniene J. The Essential Oil and Hydrolats from Myristica fragran Seeds with Magnesium Aluminometasilicate as Excipient: Antioxidant, Antibacterial, and Anti-inflammatory Activity. Foods. 2020; 9(1):37. https://doi.org/10.3390/foods9010037.

Author Response

(The authors gave the same response as above.)

Round 2

Reviewer 1 Report

The Reviewer would like to thank the Authors for addressing the review comments.